# Antinociceptive Potential of *Ximenia americana* L. Bark Extract and Caffeic Acid: Insights into Pain Modulation Pathways

**DOI:** 10.3390/ph17121671

**Published:** 2024-12-11

**Authors:** Renata Torres Pessoa, Lucas Yure Santos da Silva, Isabel Sousa Alcântara, Tarcísio Mendes Silva, Eduardo dos Santos Silva, Roger Henrique Sousa da Costa, Aparecida Barros da Silva, Jaime Ribeiro-Filho, Anita Oliveira Brito Pereira Bezerra Martins, Henrique Douglas Melo Coutinho, Jean Carlos Pereira Sousa, Andréa Rodrigues Chaves, Ricardo Neves Marreto, Irwin Rose Alencar de Menezes

**Affiliations:** 1Laboratory of Pharmacology and Molecular Chemistry, Department of Chemical Biology, Regional University of Cariri (URCA), Rua Coronel Antônio Luis 1161, Pimenta, Crato 63105-000, Ceará, Brazil; renata.pessoa@urca.br (R.T.P.); lucas.yure@urca.br (L.Y.S.d.S.); isabel.alcantara@urca.br (I.S.A.); tarcisio.mendes@urca.br (T.M.S.); eduardodos.santos@urca.br (E.d.S.S.); rogerhenrique8@hotmail.com (R.H.S.d.C.); aparecida.barros@urca.br (A.B.d.S.); jaime.ribeiro@fiocruz.br (J.R.-F.); anita.oliveira@urca.br (A.O.B.P.B.M.); 2Oswaldo Cruz Foundation (FIOCRUZ), Fiocruz Ceará, R. São José, S/N—Precabura, Eusébio 61773-270, Ceará, Brazil; 3Laboratory of Microbiology and Molecular Biology, Department of Biological Chemistry, Regional University of Cariri (URCA), Crato 63105-000, Ceará, Brazil; hdmcoutinho@gmail.com; 4Institute of Chemistry, Federal University of Goiás, Goiânia 74001-970, Goiás, Brazil; jeanpsousa01@gmail.com (J.C.P.S.); andrea_chaves@ufg.br (A.R.C.); 5Faculty of Pharmacy, Federal University of Goiás, Goiânia 74605-170, Goiás, Brazil; ricardomarreto@ufg.br

**Keywords:** natural compounds, HPLC, nociception, anti-inflammatory, mechanism of action, pain

## Abstract

**Background/Objectives:** This study evaluated the antinociceptive effect of the *Ximenia americana* L. bark extract (HEXA) and its primary component, caffeic acid (CA), through in vivo assays. **Methods**: The antinociceptive properties were assessed using abdominal writhing, hot plate, and Von Frey tests. Additionally, the study investigated the modulation of various pain signaling pathways using a pharmacological approach. **Results:** The results demonstrated that all doses of the HEXA significantly increased latency in the hot plate test, decreased the number of abdominal contortions, reduced hyperalgesia in the Von Frey test, and reduced both phases of the formalin test. Caffeic acid reduced licking time in the first phase of the formalin test at all doses, with the highest dose showing significant effects in the second phase. The HEXA potentially modulated α_2_-adrenergic (52.99%), nitric oxide (57.77%), glutamatergic (33.66%), vanilloid (39.84%), cyclic guanosine monophosphate (56.11%), and K^+^ATP channel-dependent pathways (38.70%). Conversely, CA influenced the opioid, glutamatergic (53.60%), and vanilloid (34.42%) pathways while inhibiting nitric oxide (52.99%) and cyclic guanosine monophosphate (38.98%). **Conclusions:** HEXA and CA exhibit significant antinociceptive effects due to their potential interference in multiple pain signaling pathways. While the molecular targets remain to be fully investigated, HEXA and CA demonstrate significant potential for the development of new analgesic drugs.

## 1. Introduction

Pain is a sensory and emotional experience that signals actual or potential harm to the body, thus providing an essential protective function against threats to the organism’s health [1]. Pain can be classified based on duration as acute, subchronic, or chronic [2] and based on the origin of the signaling stimulus as neuropathic or inflammatory [1].

Acute pain is immediate and prevents damage and protects injured tissue [2]. On the other hand, chronic pain lasts more than ten days. This issue is a major global health concern that impacts people in physical, economic, emotional, and social ways [3,4]. In both cases, the pain usually requires management, which is mainly performed with the use of analgesic drugs.

Nonsteroidal anti-inflammatory drugs (NSAIDs) are the most used drugs in the treatment of acute, moderate, or chronic pain [5]. Their mechanism of action involves the peripheral and central inhibition of the cyclooxygenase enzymes (COX-1 and COX-2), which inhibits the production of lipid mediators such as prostaglandins, thromboxanes, and leukotrienes [6]. However, the long-term use of NSAIDs is associated with severe adverse effects, including nausea, dyspepsia, abdominal pain, skin reactions, and nephropathy, among others [7]. In addition, NSAIDs present significant therapeutic limitations, so in many cases, the use of opioid analgesics is necessary. However, these drugs pose even more significant risks for patients. Therefore, research targeting new therapeutic alternatives with fewer adverse effects can significantly improve the pharmacological management of pain disorders [8].

In this context, medicinal plants are an ancient source of therapeutic resources in treating various diseases [9]. The pharmacological properties associated with the traditional use of medicinal species are due to the presence of bioactive compounds, indicating that medicinal plants are sources of new molecules with therapeutic potential [10] and, as such, may help develop new drugs to treat pain [11]. Historically, bark is a common ingredient that has been utilized for centuries in traditional medicine for the treatment of various diseases. This part of the plant contains different active compounds, such as tannins, salicylates, flavonoids, and alkaloids, contributing to its analgesic effects [12,13].

*Ximenia americana* L. is popularly known in Brazil as wild plum, “*ambuí*”, thorn plum, and Brazilian sandalwood [14,15]. Previous research has demonstrated that *X. americana* has antioxidant [16], gastroprotective [17], antiparasitic [18], wound healing [19], antiedematogenic [20], and antinociceptive [21] biological properties. Phytochemical studies have identified caffeic acid (CA) as a major constituent in *X. americana* [18,22,23]. Also known as 3,4-hydroxycinnamic acid, this compound is usually found in numerous fruits and vegetables, such as prunes, artichokes, eggplant, and grapes [24]. Research in pharmacology has uncovered a range of biological activities associated with CA, such as neuroprotective [25], antimicrobial [26], anticancer [27], anti-inflammatory [28], antinociception [29], and antiviral [30] activities.

Considering the ethnopharmacological evidence of *X. americana *as a promising medicinal plant, this study aimed to investigate the antinociceptive effect of *X. americana* L. bark extract (HEXA) and its main constituent, caffeic acid (CA), in animal models, to understand how these compounds can act in modulating pain induced by different stimuli.

## 2. Results

In our previous paper, we demonstrated the predominance of polyphenols in the hydroethanolic bark extract of *X. Americana* through HPLC analysis, where nine compounds were identified. Upon the new chemical approach by UPLC-MS/MS, caffeic acid and quercetin were confirmed as the major constituents (Appendix A).

The ESI-MS analysis revealed the MS/MS pattern for CA dominating the loss of a water molecule (18 Da), characterizing the fragment of *m*/*z* 161.05 ([M-H_2_O]^−^) and sequentially the decarboxylation of CO_2_ (44 Da), which can be explained by the Diels–Alder rearrangement reaction allowing the visualization of the fragment of *m*/*z* 135.05 ([M-CO_2_]^−^), as identified by [31] (Appendix A. The Molecular Networking tool with gold classification was successfully applied to correlate the fragmentation spectrum of caffeic acid and other compounds of the extract of *X. americana* L. (Appendix A) with the GNPS spectral library, allowing the confirmation of its molecular identity.

### 2.1. HEXA Failed in Inducing Central Nervous System Effects in Mice

The treatment with HEXA (200 mg/kg) did not alter the animals’ motor coordination compared to the negative control group in the rotarod test. In addition, the number of falls and the time the treated animals spent on the bar was similar to the negative control group. Concerning the open-field test, the treatment with HEXA (200 mg/kg) did not affect behavioral parameters such as rearing (lifting) and grooming (self-cleaning), nor did it alter the number of crossings. Together, these results suggest that HEXA has no evident central nervous system activity.

### 2.2. HEXA Has Antinociceptive Activity In Vivo: Central and Peripheral Effects

The administration of acetic acid induces abdominal writhing in untreated mice. treatments with HEXA at doses of 50, 100, and 200 mg/kg caused a significant reduction in the amount of abdominal writhing by 84.99%, 87.99%, and 89.49% (*p* < 0.0001), respectively, compared to the untreated group (Figure 1A).

In the hot plate model, treatments with HEXA at doses of 50, 100, and 200 mg/kg significantly increased the time the animals remained on the hot plate by 69.17%, 68.13%, and 71.98%, (*p* < 0.0001), respectively, when compared to the negative control group. A time-point analysis revealed that the lowest dose (50 mg/kg) increased the latency by 66.67%, 56.89%, 48%, and 67.89% at 30, 60, 120, and 180 min. (*p* < 0.0001). In these respective time points, the dose of 100 mg/kg increased the permanence by 68%, 50.62%, 47.32%, and 65.27%, while the dose of 200 mg/kg increased the permanence of the animals by 70.67%, 58.12%, 50%, and 72.63% (*p* < 0.0001) (Figure 1B) (F (15, 120) = 2951).

The treatments with HEXA at 50, 100, and 200 mg/kg significantly increased the pain threshold by 75.30%, 83.32%, and 74.52%, respectively, thus demonstrating the antinociceptive activity of HEXA in this model when compared to the negative control group. Regarding the time intervals, treatment with HEXA at a dose of 50 mg/kg increased the pain threshold by 42.83% (day 01), 58.83% (day 02), 69.01% (day 03), 54% (day 04), 99.39% (day 05), 90% (day 06), 51.44% (day 07), 62.43% (day 08), 84% (day 09), 77.22% (day 10), 72.49% (day 11), 76.05% (day 12), 87.46% (day 13), 93.44% (day 14), 69.67% (day 15), 64.18% (day 16), 89.42% (day 17), 89.42% (day 18), 70.50% (day 19), 74.70% (day 20), and 89.89% (day 21) (*p* < 0.0001) when compared to the negative control group (Figure 1C).

On the other hand, treatment with HEXA at a dose of 100 mg/kg increased the threshold by 44.41% (day 01), 57.95% (day 02), 69.17% (day 03), 89.24% (day 04), 92.85% (day 05), 89.64% (day 06), 99.80% (day 07), 74.90% (day 08), 89.82% (day 09), 86.30% (day 10), 87.02% (day 11), 86.57% (day 12), 99.99% (day 13), 74.75% (day 14), 77.68% (day 15), 68.94% (day 16), 93.06% (day 17), 93.06% (day 18), 96.40% (day 19), 86.31% (day 20), and 84.37% (day 21) (*p* < 0.0001), respectively, when compared to the negative control group (Figure 1C).

Finally, treatment with HEXA at a dose of 200 mg/kg increased the threshold by 45.70% (day 01), 45.92% (day 02), 53.12% (day 03), 87.85% (day 04), 97.11% (day 05), 87.09% (day 06), 93.83% (day 07), 64.70% (day 08), 71.28% (day 09), 77.22% (day 10), 61.66% (day 11), 66.50% (day 12), 99.98% (day 13), 70.66% (day 14), 60.81% (day 15), 96.24% (day 16), 83.64% (day 17), 83.64% (day 18), 99.98% (day 19), 68.62% (day 20), and 89.15% (day 21) (*p* < 0.0001), respectively, when compared to the negative control group (Figure 1C).

### 2.3. Formalin-Induced Nociception Test of HEXA and CA

Animals treated with HEXA showed a significant reduction in paw-licking time in the first phase and the second phase compared to the negative control group. In the first phase (Figure 2A), treatment with doses of 50, 100, and 200 mg/kg caused a reduction of 77.63%, 59.37%, and 58.44% (*p* < 0.001, F = 0.114), respectively, while in the second phase (Figure 2B), the same doses caused reductions of 56.74%, 69.43, and 54.02% (*p* < 0.0001, F = 15.08), respectively, compared to the control group. On the other hand, treatment with CA at doses of 0.8 and 1.8 mg/kg reduced paw-licking time by 36.40% and 22.17% (*p* < 0.0001, F = 46.26), respectively, in the first phase (Figure 2C). However, only the 1.8 mg/kg dose reduced licking time significantly by 47.97% (*p* < 0.0001, F = 23.55) in the second phase (Figure 2D) when compared to the control group.

### 2.4. Analysis of Signaling Pathways Underlying the Analgesic Effect of HEXA and CA

#### 2.4.1. Vanilloid System

The administration of HEXA (100 mg/kg), CA (1.8 mg/kg), and ruthenium red (a non-selective antagonist of transient potential receptors -TRP) significantly reduced paw-licking time by 39.84%, 34.42%, and 99.77% (*p* < 0.0001, F = 131.5), respectively, in animals challenged with capsaicin, a TRPV1 receptor agonist, when compared to the negative control group (Figure 3A). However, while the antagonist fully inhibited the nociceptive response, the natural products caused partial inhibition.

#### 2.4.2. Glutamatergic Pathway

The treatment with HEXA (100 mg/kg), CA (1.8 mg/kg), and ascorbic acid reduced the licking time by 33.66%, 53.60%, and 68.44% (*p* < 0.0001, F = 12.71) respectively, after intraplantar injection of glutamate, compared to challenged and non-treated animals. These results demonstrate the potential participation of the glutamatergic pathway in the antinociceptive effect of the compounds (Figure 3B).

#### 2.4.3. Opioid Pathway

The administration of HEXA (100 mg/kg), CA (1.8 mg/kg), and morphine significantly reduced nociception time by 57.28%, 51.21%, and 94.20%, (*p* < 0.0001, F = 26.36), respectively, when compared to the negative control group (Figure 3C). Notably, the magnitude of inhibition was significantly higher in the morphine group. When the animals received pretreatment with naloxone, the antinociceptive effect of HEXA and CA was reversed, suggesting that both treatments could modulate nociception by interfering with the opioid system.

#### 2.4.4. L-Arginine/Nitric Oxide/cGMP Pathway

Swiss mice treated with HEXA (100 mg/kg), CA (1.8 mg/kg), and L-NOARG had nociception time significantly reduced by 57.77%, 52.99%, and 51.68%, (*p* < 0.0001, F = 36.14), respectively, in comparison with the negative control group. The pretreatment with L-arginine was found to reverse the analgesic effect of HEXA and CA, suggesting an interference of HEXA and CA on the NO signaling pathway (Figure 3D).

#### 2.4.5. Cyclic Guanosine Monophosphate (cGMP) Pathway

The treatment with HEXA (100 mg/kg), CA (1.8 mg/kg), and methylene blue reduced the animals’ licking time by 56.11%, 38.98%, and 98.8% (*p* < 0.0001, F = 17.14) respectively, when compared to untreated animals. However, pretreatment with methylene blue did not affect the analgesic effect of HEXA and CA (Figure 3E). Nevertheless, the participation of this pathway in the mechanisms underlying the antinociceptive effect of the compounds cannot be ruled out.

#### 2.4.6. Involvement of the α_2_-Adrenergic Receptor

The groups of mice treated with HEXA (100 mg/kg), CA (1.8 mg/kg), and clonidine (α_2_ receptor agonist) had nociception time significantly reduced by 52.99%, 53.27%, and 92.99%, (*p* < 0.0001, F = 54.83), respectively, in comparison with the negative control group. The pretreatment with yohimbine (α_2_ receptor antagonist) was found to reverse the analgesic effect of HEXA and CA, suggesting that these products interfere with α_2_-adrenergic receptor signaling (Figure 3F).

#### 2.4.7. K+ATP Channel-Dependent Signaling

When treated with HEXA at a dose of 100 mg/kg and CA at a dose of 1.8 mg/kg, the animals showed a decrease in paw-licking time by 38.70% and 36.41% (*p* < 0.0001, F = 10.04) respectively, compared to the control group. In the HEXA group, and when the animals received pretreatment with glibenclamide, the pain-relieving effect was reversed. However, this effect was not observed in the combined treatment with glibenclamide and CA (see Figure 3G), suggesting that HEXA components could partially act on K+ATP channels. However, this effect was not observed in the combined treatment with glibenclamide and CA (Figure 3G).

#### 2.4.8. Cholinergic Pathway

The treatment with HEXA (100 mg/kg), CA (1.8 mg/kg), and acetylcholine reduced nociception time by 53.33%, 39.55%, and 62.67% (*p* < 0.0001, F = 17.15), respectively, when compared to the control group. When the animals in the HEXA and CA groups received pretreatment with atropine, the antinociceptive effect was not affected, indicating that there is no clear relationship between this mechanism and the analgesic effect of the natural products investigated in this study (Figure 4A).

#### 2.4.9. Adenosinergic Pathway

The treatment with HEXA (100 mg/kg) and CA (1.8 mg/kg) significantly reduced the animals’ licking time by 49.27% and 59.12% (*p* < 0.0001, F = 15.37), respectively, after intraplantar formalin injection compared to the control group. However, there was no statistical difference when the animals in the HEXA and CA groups received pretreatment with caffeine, a non-specific inhibitor of the pathway (see Figure 4B).

#### 2.4.10. Dopaminergic System

The treatment with HEXA (100 mg/kg) and CA (1.8 mg/kg) significantly reduced by 41.87% and 49.49%, (*p* < 0.0001, F = 23.70), respectively, the animals’ licking time after when compared to the negative control group. However, no statistical difference was observed when the animals of the HEXA and CA groups received pretreatment with haloperidol, thus suggesting that none of these substances inhibit the adenosinergic signaling pathway (Figure 4C).

#### 2.4.11. Involvement of the Serotonergic System

After receiving HEXA (100 mg/kg) and CA (1.8 mg/kg) treatments, the animals showed a 46.65% and 45.73% (*p* < 0.0001, F = 15.33) reduction in licking time compared to the control group. Additionally, when the HEXA and CA groups were given a pre-treatment of PCPA, the antinociceptive effect was sustained, indicating that both HEXA and CA do not affect the serotonergic signaling pathway (Figure 4D).

## 3. Discussion

This study contributes new insights into the potential mechanisms of action underlying the antinociceptive effect of *Ximenia americana* L. bark extract (HEXA) and its major constituent caffeic acid (CA). In this context, significant reductions in pain responses were demonstrated through different assessment tests such as formalin, acetic acid, hot plate, and complete Freund’s adjuvant as well as through the pharmacological modulation of pain signaling pathways that to date have not been previously reported in the literature. Previous studies have shown that HEXA has clinical safety by presenting low toxicity (LD_50_ > 2000 mg/kg), as well as caffeic acid (LD_50_ = 4850 mg/kg) [32,33].

In our previously published data, our research group demonstrated that the extract of *Ximenia americana* L. used in this study (HEXA) has a unique chemical profile. It is characterized by the presence of alkaloids, steroids, saponins, glycosides, cyanogenic glycosides, anthraquinone, and, principally, flavonoids, terpenoids, and tannin derivatives. The quantitative analysis by HPLC revealed rutin, gallic acid, quercetin, catechin, kaempferol, chlorogenic acid, ellagic acid, caffeic acid, and quercitrin as the main compounds [18,22,23,34]. The literature’s comprehensive chemical profile, combined with the isolation of several other compounds from this species, including sambunigrin, gallic acid, catechin, and different stereoisomers of (epi)catechin, gallotannins quercetin, quercetin-3-O-β-xylopyranoside, quercetin-3-O-(6″-galloyl)-β-glucopyranoside quercitrin, avicularin, β-glucogalin, 1,6-digalloyl-β-glucopyranose, and kaempferol-3-O-(6″-galloyl)-β-glucopyranoside [16,35,36,37,38,39], underscores its potential for drug discovery, providing a solid foundation for further research in pharmacology.

Previous research by Coutinho et al. (2009) demonstrated that quercitrin presents anti-inflammatory effects, while epicatechin isolated from ethanolic extracts of *X. americana* bark [40,41] and flavonoid-rich fractions obtained from the roots of this species showed antinociceptive activity and antioxidant potential [42]. In addition, caffeic acid has previously been shown to have biological activities such as antimicrobial [26], anticancer [43], muscle relaxant [44], antioxidant [45], gastroprotective, and antinociceptive activities. Corroborating with our results, the literature shows evidence that this antinociceptive effect is dose-dependent and may be mediated by nitric oxide [46] through the downregulation of NF-κB, α_2_-adrenergic and opioidergic receptors [29], but it does not involve serotonergic pathways.

Thus, the antinociceptive activity of the HEXA can be justified by the presence of bioactive compounds with proven pharmacological properties. Our studies demonstrated the therapeutic potential of various formulations obtained from different parts of *X. americana*. For instance, an extract from the stem of this species showed topical anti-inflammatory potential in ear edema models [22], while the ethanol extract showed anticonvulsant activity in models of acute inflammation [47]. Moreover, the total polysaccharides obtained from *X. americana* showed anti-inflammatory and antinociceptive action in a model of acute pancreatitis. It was demonstrated that their pharmacological actions might occur through type 2 cannabinoid receptors [48].

Regarding the biological activities investigated in the present study, it was observed through the open-field and rotarod tests that HEXA, at the highest dose evaluated in the present study, did not promote central nervous system activity, suggesting that the antinociceptive effects observed in this study do not occur due to a change in behavioral parameters such as incoordination from either stimulant or depressant, evaluated in the rotarod and open-field models. The study found that HEXA, when orally administered at doses of 50, 100, and 200 mg/kg, demonstrated an antinociceptive effect by reducing nociception in mice models challenged with acetic acid and formalin, as evidenced by decreased abdominal writhing and paw licking. The results of the study showed that HEXA significantly reduced the amount of writhing, indicating a possible peripheral analgesic effect by inhibiting the release of these mediators.

Regarding the formalin test, treatment with HEXA and CA significantly reduced nociception in both the first and second phases of the test, suggesting a possible inhibition of cell activation and mediator release. Zhen et al. (2015) demonstrated that an extract obtained from *X. caffra*, chemically characterized by the presence of phenolic compounds also identified in *X. americana*, including gallic acid, catechin, quercetin, and kaempferol and its derivatives, showed antioxidant, antiproliferative, and anti-inflammatory potential, reinforcing the potential action of the extract on the inflammatory component of the nociceptive response [49]. In the hot plate test, the HEXA significantly increased the time the animals remained on the plate, which suggests the intriguing possibility that the extract may also act at a supraspinal level.

Complete Freund’s adjuvant (CFA) is commonly used in chronic in vivo models. It triggers an inflammatory process that lowers nociceptive thresholds, leading to the development of allodynia and hyperalgesia in inflamed limbs. This compound induces an inflammatory response involving mediators such as histamine, serotonin, cytokines, glutamate, and prostaglandins [50,51], resulting in edema and persistent mechanical hyperalgesia in chronic models [52], as well as causing long-lasting mechanical allodynia in mice [53]. The latest findings demonstrate that HEXA significantly reduces allodynia and hyperalgesia, supporting the idea that this extract might inhibit the release of nociceptive process mediators, as previously suggested in this study. A study by Gamato et al. (2011) revealed that caffeic acid was more potent than its derivative rosmarinic acid in nociception models, with the 10 mg/kg dose showing the most significant antinociceptive potential [54].

The extract of *X. americana* stem significantly reduced the chronic inflammatory process in a tendinitis model [55]. The study conducted by Da Palma et al. (2020) demonstrated that hydroalcoholic extracts obtained from the leaves and bark of *X. americana* presented wound-healing effects in acute, subchronic, and chronic stages of injury in mice. Thus, the therapeutic potential of this species in treating acute pain in both the acute and chronic phases is suggested. Given these results, the present study aimed to investigate the potential signaling pathways underlying the analgesic effects of HEXA and its major constituent caffeic acid. For this purpose, we chose signaling pathways that are critically involved in the painful process, including the opioid, cholinergic, α2-adrenergic, adenosinergic, dopaminergic, glutamatergic, vanilloid, and serotoninergic pathways as well as those mediated by cyclic guanosine monophosphate, nitric oxide, and K+ATP channels.

The opioid signaling pathway involves G protein-coupled receptors (Gi/0 subfamily), whose activation inhibits the enzyme adenylate cyclase, leading to the degradation of cyclic adenosine monophosphate (cAMP) and blocking downstream events mediated by protein kinase A (PKA) [56]. Moreover, through this signaling pathway, opioid agonists open potassium channels and inhibit the opening of voltage-gated calcium channels [57], reducing the activity of nociceptor neurons and the release of neurotransmitters [56]. The results obtained in this study indicate that HEXA and CA can interfere with the opioid signaling pathway, which corroborates the results obtained in the hot plate assay.

Nitric oxide (NO) indirectly mediates the inflammatory process, contributing to vascular and cellular events. Since this mediator is produced from L-arginine under the action of NO-synthase (NOS), the inhibition of this synthesis pathway is associated with the inhibition of the pain process [58,59]. L-NOArg, an L-arginine analog, competitively inhibits NOS, blocking thus nitric oxide synthesis [60]. The present study showed that both HEXA and CA affect NO-mediated pain, corroborating the study by Kolgazi et al. (2021), who suggested that the gastroprotective potential of caffeic acid depends on the modulation of the nitric oxide pathway by regulation of NF-κB through NIK/IKK and c-Src/ERK signaling pathways [61].

The vanilloid receptor pathway or transient receptor potential vanilloid type 1 (TRPV1) is sensitive to a variety of harmful stimuli, such as temperatures elevated to ±42 °C, lipid mediators, and vanilloid compounds such as capsaicin and resiniferatoxin [62,63]. The activation of these receptors results in neuron depolarization, leading to the release of multiple neuropeptides [64]. Ruthenium red, a non-selective antagonist of vanilloid receptors, is used to evaluate the potential interaction of substances with vanilloid receptors [65]. Glutamate induces a nociceptive response by activating glutamatergic receptors at supraspinal, spinal, and peripheral sites [66]. This neurotransmitter activates AMPA (α-amino-3-hydroxy-5-methyl-4-isoxazole propionic acid) and NMDA (N-Methyl-D-Aspartic acid) receptors, thus allowing the passage of Na^+^ and Ca^2+^, facilitating the action potential and subsequent nociceptive responses [67].

Such inhibition also results in the availability of cGMP, providing peripheral antinociception through the opening of ATP-sensitive K^+^ channels [68]. On the other hand, the administration of methylene blue causes the activation of the guanylate cyclase enzyme, which catalyzes cGMP production [69]. In the present study, it was evidenced that both HEXA and CA act synergistically. The analgesic pathway mediated by α_2_ receptors involves Gi-mediated inhibition of presynaptic Ca^2+^ channels, preventing glutamate (excitatory neurotransmitter) from being released from primary afferent fibers [70]. Simultaneously, the opening of K^+^ channels results in hyperpolarization, causing a decrease in the excitability of postsynaptic cells [71]. Agonists of these receptors have analgesic properties involving peripheral and central mechanisms [72]. In this study, the evaluation of HEXA in this pathway indicates that the antinociceptive effect of this extract can involve the participation of these receptors.

K+ATP channels are directly involved in cell hyperpolarization, consecutively decreasing intracellular concentrations of Ca^2+^ and reducing the release of neurotransmitters, thus inhibiting nociception. Experimentally, the administration of glibenclamide blocks these channels [73]. From the results obtained in the study, it is reasonable to hypothesize that the action of HEXA affects the activation of potassium channels, contributing to the observed analgesic effect.

The findings of this study indicate that HEXA exhibits antinociceptive effects both centrally and peripherally. These effects may be attributed to its ability to suppress the release of inflammatory mediators or regulate their interaction with receptors involved in pain perception. This research revealed that the components of the extract of *X. americana* can influence multiple pain pathways with the modulation of α2-adrenergic, nitric oxide, glutamatergic, vanilloid, cyclic guanosine monophosphate, and K+ATP channel-dependent pathways. On the other hand, caffeic acid specifically influences the opioid, glutamatergic, and vanilloid pathways by inhibiting nitric oxide and cyclic guanosine monophosphate.

## 4. Materials and Methods

### 4.1. Drugs, Reagents, and Doses

The substances used in this study were obtained from Sigma-Aldrich. The other substances used in the tests were the following: formalin (PubChem CID = 712), morphine (5 mg/kg; s.c.) (PubChem CID = 5288826), L-NOARG ((PubChem CID = 440005), ruthenium red (PubChem CID = 117587625), capsaicin (PubChem CID = 1548943), glutamate (PubChem CID = 23689119), methylene blue (PubChem CID = 6099), clonidine (PubChem CID = 20179), yohimbine (PubChem CID = 6169), glibenclamide (PubChem CID = 3488), acetylcholine (PubChem CID = 6060), atropine (PubChem CID = 174174), caffeine (PubChem CID = 2519), and p-chlorophenyl alanine (PubChem CID = 46520). Methanol, acetonitrile (LiChrosolv^®^, ≥99.9%), and ammonium hydroxide (NH4OH) were purchased from Sigma-Aldrich (St. Louis, MO, USA).

HEXA was obtained as previously described by da Silva et al. (2018) [20] and administered at 50, 100, and 200 mg/kg orally. Caffeic acid (PubChem CID = 689043) was administered at 0.18 mg/kg and 1.8 mg/kg.

### 4.2. Chemical Profile by Electrospray Ionization–Mass Spectrometry (ESI-MS) Analysis

In the previous work, Silva et al. (2018), we determined the chemical profile of HEXA compounds, identifying 21 compounds using HPLC-DAD. However, in the current study, the analysis focuses specifically on the compound caffeic acid. For this purpose, electrospray ionization–mass spectrometry (ESI-MS) analysis was utilized.

The sample was prepared by dissolving the dried extract in 1 mL of a methanol/acetonitrile mixture (2:1, *v:v*) containing 10 μL of ammonium hydroxide. The ESI-MS analyses were performed on a high-resolution Quadrupole-Orbitrap hybrid mass spectrometer (Q-Exactive, Thermo Scientific, Waltham, MA, USA). The ESI source conditions were set as follows: negative ionization mode, spray voltage of 3.2 kV, capillary temperature of 320 °C, S-Lens RF level at 45 V, sheath gas flow rate at 8 L min^−1^, and auxiliary gas flow rate at 5 L.min^−1^, and for MSMS, a collision energy of 15 eV was used.

#### Compound Annotation

The mass spectrometer data files were converted from (.raw) to (.mzML) format using MSconvert software v.3 (ProteoWizard, Palo Alto, CA, USA) and processed using the online platform (https://gnps.ucsd.edu/ProteoSAFe/static/gnps-splash.jsp, accessed on 10 July 2024) to retrieve compound information. Molecular annotations were based on the comparison of experimental MS/MS spectra with the GNPS spectral library using tools such as Molecular Networking [74].

### 4.3. Animals

The animals used were Swiss mice (*Mus musculus*) of both sexes weighing 20–30 g and housed in polypropylene cages (410 × 282 × 150 mm; n = 6) at 24 ± 2 °C with a 12 h light/dark cycle, with free access to water and specific chow (Labina, Presence^®^). Mice were fasted from solids for 6–8 h before experiments.

### 4.4. Ethical Information

The study adhered to guidelines from CONCEA, U.K. Animals (Scientific Procedures) Act, 1986, EU Directive 2010/63/EU, and NIH guidelines. Protocols were approved by the Ethics Committee on Animal Use of URCA (approval number 180/2020.2).

### 4.5. Evaluation of the Effects of HEXA on the Central Nervous System

#### 4.5.1. Rotarod Performance Test

Mice (n = 3/group) were treated with water (10 mL/kg; p.o.) or HEXA (200 mg/kg; p.o.). After 1 h, the number of falls and time on the rotating rod were recorded [75].

#### 4.5.2. Open-Field Test

Mice (n = 3/group) were treated with water (10 mL/kg; p.o.) or HEXA (200 mg/kg; p.o.). After 1 h, the number of fields explored in a 30 cm^2^ open field was observed for 5 min [76].

### 4.6. Screening for the Antinociceptive Effect

The antinociceptive effect was assessed using acetic acid-induced writhing, formalin, hot plate, and Von Frey tests. Mice were divided into groups (n = 6) and treated with water (10 mL/kg; p.o.) or HEXA (50, 100, or 200 mg/kg; p.o.) [77]. The experimental timeline is shown in Figure 1.

#### 4.6.1. Acetic Acid-Induced Abdominal Contortions

Mice (n = 6/group) were treated with water (10 mL/kg; p.o.) or HEXA (50, 100, or 200 mg/kg; p.o.) 1 h before 0.6% acetic acid (i.p.). The amount of writhing was recorded for 30 min [76,78].

#### 4.6.2. The Formalin Test

Mice (n = 6/group) were treated with water (10 mL/kg; p.o.), HEXA (50, 100, or 200 mg/kg; p.o.), or CA (0.18, 1.8 mg/kg). After 1 h, 2.5% formalin (20 μL) was administered, and the nociceptive response was observed at 0–5 min and 15–30 min post injection [79].

### 4.7. Evaluation of Central and Peripheral Antinociceptive Responses

#### 4.7.1. Hot Plate Test

Mice (n = 6/group) were treated with water (10 mL/kg; p.o.) or HEXA (50, 100, or 200 mg/kg; p.o.). Latency time was recorded at various intervals up to 180 min post treatment [76,80].

#### 4.7.2. Mechanical Pressure Hypernociception (Von Frey Test)

Mice (n = 6/group) were subjected to mechanical stimulation in the right paw at 24 h and 30 min (basal measure) before the test. After 30 min, the animals received an injection (20 μL/paw) of complete Freund’s adjuvant (CFA) (1 mg/mL) in the right hind paw. The animals were treated with water (10 mL/kg; p.o.) or HEXA (50, 100, or 200 mg/kg, p.o.). Subsequently, treatments were carried out on the 5th, 9th, 13th, 17th, and 21st days after induction. The test was performed using the Von Frey apparatus. Briefly, the animals were placed individually in glass boxes (12 × 20 × 20 cm) supported on an elevated surface and covered with a wire mesh. After 1 h of acclimatization, the stimulus was applied with a rod with a constant force on the plantar surface of the right hind paw (to assess hypernociception to the lesion in injured mice) using the Von Frey filament resistance apparatus for 21 consecutive days. The force in (g) was determined by removing the paw from the filament. The force was applied six times, and the average of the three most similar values was considered [81,82].

### 4.8. Investigation of the Signaling Pathways Associated with the Analgesic Effect of HEXA and AC

The effect of HEXA (100 mg/kg; b.w.) or CA (1.8 mg/kg; b.w.) on pain signaling pathways was evaluated using the formalin assay through the pharmacological signaling pathways, including opioid receptor, nitric oxide, vanilloid, glutamatergic, cyclic guanosine monophosphate pathway, adrenergic α-2, K+ATP channels, cholinergic, adenosinergic, dopaminergic, and the serotonergic pathways.

In the formalin model, mice (n = 6) were pretreated with HEXA (100 mg/kg; p.o.), CA (1.8 mg/kg; p.o.), or, according to the signaling pathway: opioid—morphine (agonist—5 mg/kg; s.c.) and naloxone (antagonist—4 mg/kg; i.p.) [83]; nitric oxide—L-NOARG (nitric oxide synthase (NOS) inhibitor—75 mg/kg; i.p.) and L-arginine (NOS substrate—600 mg/kg; i.p.) [84,85]; vanilloid—ruthenium red (non-selective TRP antagonist—3 mg/kg; i.p.) and capsaicin (TRPV1 receptor agonist—5.2 nmol/paw) [66]; glutamatergic—ascorbic acid (100 mg/kg; i.p.) and buffered glutamate (20 μmol/paw) [86,87,88,89]; cGMP—methylene blue (guanylate cyclase inhibitor—20 mg/kg; i.p.) [66]; α_2_-adrenergic—clonidine (agonist—0.1 mg/kg; i.p.) and yohimbine (antagonist—0.15 mg/kg; i.p.) [90]; K+ATP channels—glibenclamide (K+ATP channel blocker—3 mg/kg; i.p.) [66,91]; cholinergic—acetylcholine (agonist—1 mg/kg; i.p.) and atropine (antagonist—1 mg/kg; i.p.) [85]; adenosinergic—caffeine (10 mg/kg; i.p.) [92]; dopaminergic—haloperidol (non-selective dopamine receptor antagonist—0.2 mg/kg; i.p.) [93,94]; and serotoninergic—p-chlorophenyl alanine (PCPA) (serotonin receptor antagonist—100 mg/kg; i.p.) [95,96].

### 4.9. Statistical Analysis

The values were expressed as mean ± SEM. Data were analyzed by one-way ANOVA followed by Tukey’s test or two-way ANOVA followed by Dunnett’s or Tukey’s post hoc, using the GraphPad Software version 6.1. *p*-values < 0.05 were considered significant.

## 5. Conclusions

In conclusion, HEXA and CA have significant antinociceptive effects in the absence of evident central nervous system (CNS) toxicity in mice and, as such, have the potential to be used in the development of new analgesic drugs. The present results bring new insights into the antinociceptive effects of HEXA and CA as well as the main mechanisms underlying their effects in pain management, shedding light on the search for new therapeutic alternatives.

## Figures and Tables

**Figure 1 pharmaceuticals-17-01671-f001:**
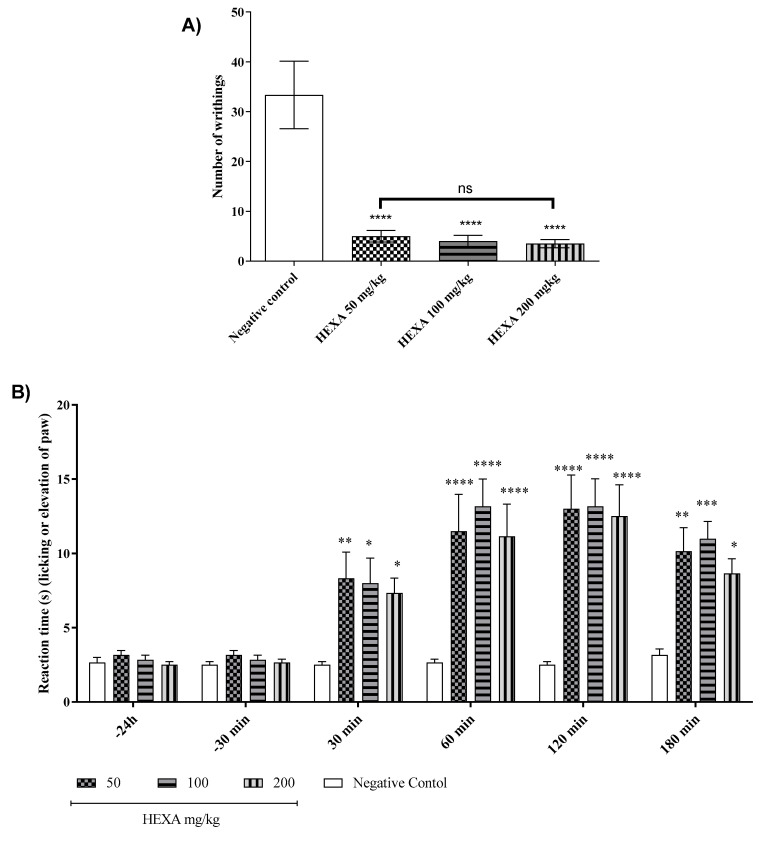
Antinociceptive effect of HEXA (50, 100, and 200 mg/kg) by acetic acid-induced abdominal writhing test (**A**); the hot plate test (**B**) and the hypernociception measure by Von Frey test induced by CFA (**C**). The arrow indicates the treatment day. One-way ANOVA followed by Tukey’s test. (* *p* < 0.05; ** *p* < 0.01; *** *p* < 0.001; **** *p* < 0.0001; ns = not significant when compared to the negative control group) for the acetic acid-induced abdominal writhing test. Two-way ANOVA followed by Tukey’s test (* *p* < 0.05; ** *p* < 0.01; *** *p* < 0.001; **** *p* < 0.0001 when compared to the negative control group) for hot plate and hypernociception induced by CFA assay.

**Figure 2 pharmaceuticals-17-01671-f002:**
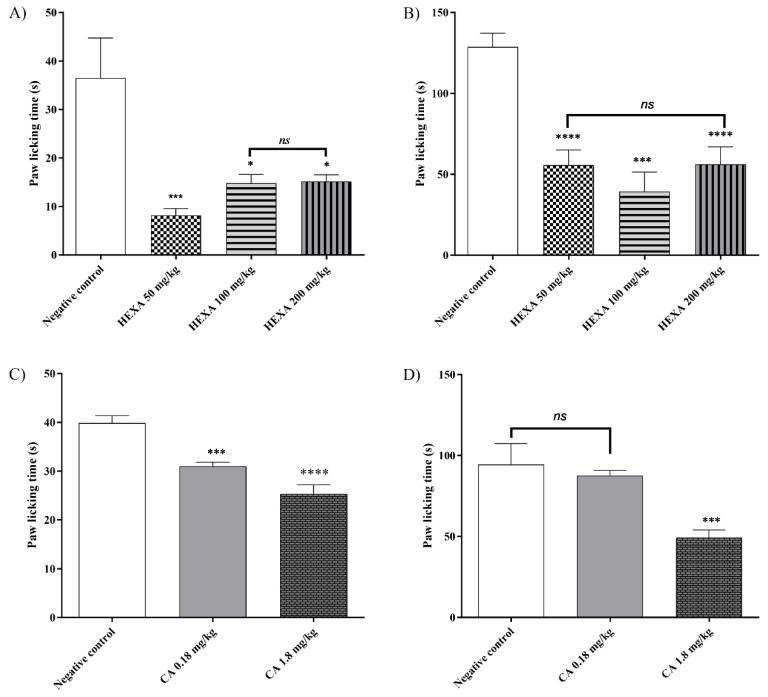
Evaluation of the antinociceptive effect of HEXA (50, 100, and 200 mg/kg) and CA (0.18 and 1.8 mg/kg) on the neurogenic phase (phase 1—(**A**,**C**)) and inflammatory phase (phase 2—(**B**,**D**)) against formalin-induced pain in mice. These values represent the arithmetic mean ± SE (Standard Error of the Mean) (n = 6/group). One-way ANOVA followed by Tukey’s test. (* *p* < 0.05; *** *p* < 0.001; **** *p* < 0.0001; ns = not significant when compared to the negative control group).

**Figure 3 pharmaceuticals-17-01671-f003:**
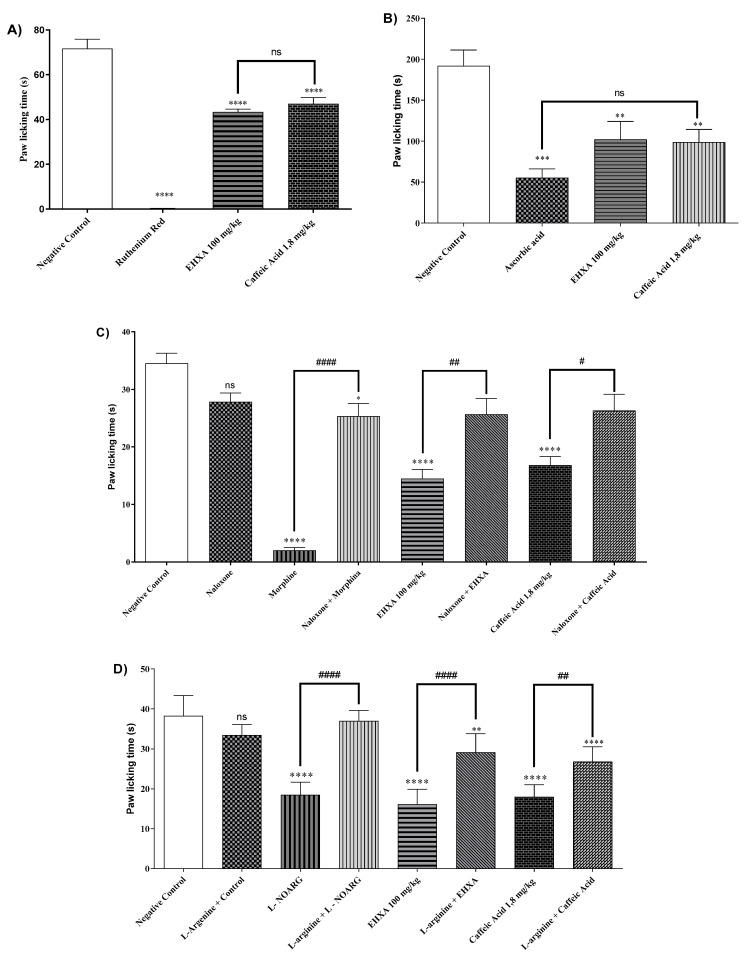
Signaling pathways underlying the antinociceptive response of HEXA (100 mg/kg) and CA (1.8 mg/kg) in the antinociceptive response: (**A**) vanilloid; (**B**) glutamatergic; (**C**) opioid; (**D**) L-Arginine/Nitric Oxide/cGMP; (**E**) cyclic guanosine monophosphate. (**F**) Participation of α2-adrenergic receptors; (**G**) K+ATP channels against formalin-induced pain in mice. Values present the mean ± SE (Standard Error of the Mean) (n = 6/group). One-way (ANOVA) followed by the Tukey test (* *p* < 0.05; ** *p* < 0.01; *** *p* < 0.001; **** *p* < 0.0001 when compared to the negative control group; # *p* < 0.05; ## *p* < 0.01; #### *p* < 0.0001 when comparing agonist vs. antagonist + agonist or HEXA alone vs. antagonist + HEXA group; ns = not significant).

**Figure 4 pharmaceuticals-17-01671-f004:**
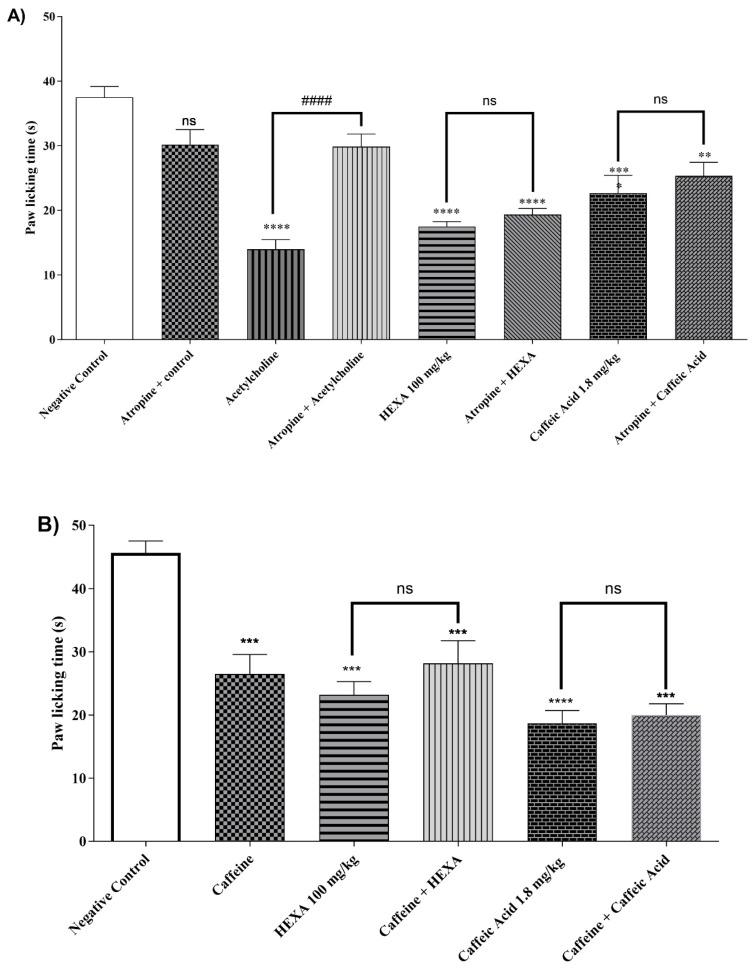
Participation of (**A**) cholinergic, (**B**) adenosinergic, (**C**) dopaminergic pathway, and (**D**) serotonergic system for the antinociceptive response of HEXA (100 mg/kg) and CA (1.8 mg/kg) against formalin-induced pain in mice. The values present the mean ± SE (Standard Error of the Mean) (n = 6/group). One-way (ANOVA) followed by the Tukey test (* *p* < 0.05; ** *p* < 0.01, *** *p* < 0.001; **** *p* < 0.0001 when compared to the negative control group; #### *p* < 0.0001 when comparing agonist vs. agonist + antagonist; ns = not significant).

## Data Availability

Data is contained within the article or Appendix A.

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
