# Peer review of "Antinociceptive Potential of Ximenia americana L. Bark Extract and Caffeic Acid: Insights into Pain Modulation Pathways"

_pharmaceuticals, 2024, doi:10.3390/ph17121671_

Round 1
Reviewer 1 Report
Comments and Suggestions for Authors
This manuscript on Ximenia americana L. highlights the significant pain-relieving effects of its bark extract and main component, caffeic acid. Using in vivo tests, the research showed that bark extract and caffeic acid effectively reduced pain behaviors and interacted with multiple pain pathways, including the nitric oxide and glutamatergic systems. Both compounds demonstrated potential for future analgesic drug development, though further research is needed to fully understand their molecular targets.
The abstract is too long and not very clear. You try to give all the information with the results in the abstract. Please revise – make it shorter and just highlight the main purpose and the main findings in the manuscript.
The authors have performed variety of in vivo assays, including abdominal writhing, hot plate, and Von Frey tests, in order to assess the antinociceptive effects. Also, another strength of the article is the exploration of multiple pain signaling pathways providing valuable information about the mechanisms through which HEXA and CA exert their effects.
In the results section you do not need to show the mas spectra. Please add them to a supplementary file and cite them.
I have a few questions for the authors, which they can answer and include in the manuscript.
Can you elaborate on the specific molecular mechanisms through which HEXA and caffeic acid modulate the various pain pathways mentioned in your study?
How did you determine the optimal doses of HEXA and caffeic acid for your experiments, and what criteria were used to select these specific concentrations?
What are the potential long-term effects of HEXA and caffeic acid on pain modulation, and have you conducted chronic pain studies to assess their sustained efficacy?
Were there any observed side effects associated with the administration of HEXA or caffeic acid in your animal models, particularly concerning motor coordination or behavior?
Regarding my comments, I suggest a minor revision of the proposed manuscript.
Author Response
The abstract is too long and not very clear. You try to give all the information with the results in the abstract. Please revise – make it shorter and just highlight the main purpose and the main findings in the manuscript.
Thanks for the comment. Corrections were made to the manuscript.
The authors have performed a variety of in vivo assays, including abdominal writhing, hot plate, and Von Frey tests, in order to assess the antinociceptive effects. Also, another strength of the article is the exploration of multiple pain signaling pathways, providing valuable information about the mechanisms through which HEXA and CA exert their effects.
Response: Thanks for the comment.
In the results section, you do not need to show the mas spectra. Please add them to a supplementary file and cite them.
Response: Thanks for the comment. Mass spectra have been moved to the supplementary material.
I have a few questions for the authors, which they can answer and include in the manuscript.
Can you elaborate on the specific molecular mechanisms through which HEXA and caffeic acid modulate the various pain pathways mentioned in your study?
Response: Thank you for your comment. Due to the large volume of data in the article, we will explore biochemical or molecular assays in future studies.
How did you determine the optimal doses of HEXA and caffeic acid for your experiments, and what criteria were used to select these specific concentrations?
Response: Thank you for your comment. Initially, the LD50 screening was conducted to determine the maximum dose of the extract that does not produce toxic effects, for the caffeic acid we utilized the LD50 value reported in the literature or concentration correspondent identified by HPLC-DAD analysis. For the HEXA extract, following this procedure, dose screening was performed using 10% of the maximum concentration that showed no toxic effect. Sequential doses were then prepared by diluting in a 1:2 proportion. The most effective dose was identified through tests such as the acetic acid-induced writhing test, formalin test, hot plate test, and Von Frey test was subsequently used in follow-up studies to investigate the potential pain signaling pathways involved.
What are the potential long-term effects of HEXA and caffeic acid on pain modulation, and have you conducted chronic pain studies to assess their sustained efficacy?
Response: Thank you for the comment. The subchorionic hypernociception assay was evaluated using Complete Freund's Adjuvant (CFA). In this assay, hypernociception persisted for 21 days. The results demonstrated the efficacy of both substances.
Were there any observed side effects associated with the administration of HEXA or caffeic acid in your animal models, particularly concerning motor coordination or behavior?
Response: Thank you for the comment. Based on the evaluation of the open field and Rotarod assays, both substances, HEXA and caffeic acid, at their maximal doses, did not show any effects on the central nervous system or impair motor or behavioral coordination. These findings are consistent with those reported in the literature.
Reviewer 2 Report
Comments and Suggestions for Authors
I rate the works highly in terms of their content and clarity of message. The research is a continuation of previous studies, it broadens knowledge and explains many problems. The authors have presented and discussed the results of their research in a clear and legible manner. Minor comments and suggestions have been noted in the manuscript. Congratulations on your good work.

Author Response
Minor comments and suggestions have been noted in the manuscript. Congratulations on your good work
Response: Thanks for the comment. All corrections indicated have been made.
Reviewer 3 Report
Comments and Suggestions for Authors
I am grateful for being invited to review the manuscript “Antinociceptive Potential of Ximenia americana L. Bark Extract and Caffeic Acid: Insights into Pain Modulation Pathways,” submitted by Renata Torres Pessoa and colleagues for publication in the pharmaceuticals journal.
The study investigates the antinociceptive potential of Ximenia americana bark extract (HEXA) and its primary component, caffeic acid (CA), using multiple in vivo pain models. HEXA and CA were tested for their ability to modulate pain via various signaling pathways, with results indicating both possess significant analgesic activity. HEXA showed broad-spectrum effects, influencing multiple pathways, while CA primarily affected the opioid, glutamatergic, and vanilloid pathways. These findings highlight the therapeutic potential of HEXA and CA as candidates for analgesic drug development.
The study is grounded in ethnopharmacological evidence supporting the use of Ximenia americana in traditional medicine.
The study uses various models (abdominal writhing, hot plate, Von Frey, and formalin tests) to provide a comprehensive assessment of antinociceptive activity.
It explores multiple pain modulation pathways, such as opioid, glutamatergic, and vanilloid systems, enhancing understanding of the molecular mechanisms involved.
The findings indicate potential applications for HEXA and CA in developing alternative analgesic agents, addressing the need for new pain therapies.
I have the following suggestions to improve the manuscript further:
1.While the study provides pharmacological insights, it lacks molecular or biochemical validation (e.g., receptor binding assays or genetic studies) to confirm the involvement of proposed pathways.
2.The effects of different doses of HEXA and CA are mentioned but not thoroughly analyzed, limiting the understanding of their pharmacodynamics.
3.The differentiation between HEXA and CA’s mechanisms is insufficiently detailed, leaving potential synergies or redundancies unclear.
4.Safety profiles of HEXA and CA are not addressed, which is critical for drug development.
5."...to treat diseases such as diarrhea, fever, wounds, and pain." The use of "diseases" is inappropriate for conditions like pain or wounds. Replace with "conditions" or "ailments."
6."Additionally, the study investigated the modulation of various pain signaling pathways using a pharmacological approach." Vague phrasing of "pharmacological approach." Specify the methods (e.g., receptor antagonists or inhibitors).
7."...and inhibited both phases of the formalin test." "Inhibited" might not be precise for describing behavioral changes. Use "reduced" instead.
8.The abstract should include quantitative data from the results to strengthen its impact. For example, specify the percentage reduction in abdominal writhing, latency increases in the hot plate test, or the extent of pathway modulation. This will make the findings more precise and compelling for readers.
Author Response
I have the following suggestions to improve the manuscript further:
1.While the study provides pharmacological insights, it lacks molecular or biochemical validation (e.g., receptor binding assays or genetic studies) to confirm the involvement of proposed pathways.
Response: Thank you for the comments and suggestions. However, due to the extensive volume of data presented in this manuscript, these procedures will be conducted in future studies to explore further the findings related to the signaling pathways.
2.The effects of different doses of HEXA and CA are mentioned but not thoroughly analyzed, limiting the understanding of their pharmacodynamics.
Response: Thanks for your comments. The paper's aim was first to focus on the effects (pharmacodynamics) and efficacy. Future studies will complement these findings by examining molecular or biochemical evidence correlated with the signaling pathways.
3.The differentiation between HEXA and CA’s mechanisms is insufficiently detailed, leaving potential synergies or redundancies unclear.
Response: Thank you for your comment. We did not study the possibility of synergistic or redundant effects. The purpose of studying caffeic acid was to validate whether part of the antinociceptive effect observed for the extract was related to the action of this phenolic acid.
4.Safety profiles of HEXA and CA are not addressed, which is critical for drug development.
Response: Thanks for the comment. The acute toxic assay showed that the extract and caffeic acid are safety for use based on the LD50. The LD50 value of the same extract has previously been published in a study by the group (Pessoa, R.T.; Alcântara, I.S.; da Silva, L.Y.S.; da Costa, R.H.S.; Silva, T.M.; de Morais Oliveira-Tintino, C.D.; Ramos, A.G.B.; de Oliveira, M.R.C.; Martins, A.O.B.P.B.; de Lacerda, B.C.G.V.; et al. Ximenia Americana L.: Chemical Characterization and Gastroprotective Effect. Analytica 2023, 4, 141–158, doi:10.3390/analytica4020012) and LD50 of caffeic acid (Liu, Y., Qiu, S., Wang, L., Zhang, N., Shi, Y., Zhou, H., Liu, X., Shao, L., Liu, X., Chen, J., & Hou, M. (2019). Reproductive and developmental toxicity study of caffeic acid in mice. Toxicologia QuíMica e Alimentar, 123, 106-112. https://doi.org/10.1016/j.fct.2018.10.040) has already reported its toxicity in the literature.
5."...to treat diseases such as diarrhea, fever, wounds, and pain." The use of "diseases" is inappropriate for conditions like pain or wounds. Replace with "conditions" or "ailments."
Response: Done, thank you for the comment.
6."Additionally, the study investigated the modulation of various pain signaling pathways using a pharmacological approach." Vague phrasing of "pharmacological approach." Specify the methods (e.g., receptor antagonists or inhibitors).
Response: Thanks for the comment. From a pharmacological point of view, associations between antagonists and agonists are used in order to evaluate the effect/reversal of the tested substance, which are described in more detail in section 4 in material and methods.
7."...and inhibited both phases of the formalin test." "Inhibited" might not be precise for describing behavioral changes. Use "reduced" instead.
Response: Done, thank you for the comment.
8.The abstract should include quantitative data from the results to strengthen its impact. For example, specify the percentage reduction in abdominal writhing, latency increases in the hot plate test, or the extent of pathway modulation. This will make the findings more precise and compelling for readers.
Response: Thanks for the comment. Corrections were made to the manuscript.
Reviewer 4 Report
Comments and Suggestions for Authors
I have thoroughly reviewed the manuscript titled "Antinociceptive Potential of Ximenia americana L. Bark Extract and Caffeic Acid: Insights into Pain Modulation Pathways" in detail and found it quite interesting. I find this manuscript intriguing, particularly in its exploration of antinociceptive activity through various experimental models and its investigation into the potential mechanisms underlying this activity. I think the manuscript has good potential for acceptance in Pharmaceuticals. However, the researchers are advised to address the following questions and consider the recommendations provided.
1. The scientific names used in the manuscript should adhere to the proper conventions for writing scientific names as per international standards. For example, the genus and specific epithet should be italicized, the complete scientific name should include the author's name, and when mentioned subsequently, the genus name should be abbreviated to its initial, capitalized, and followed by a full stop. In this manuscript, some instances of scientific names do not comply with these conventions. Therefore, the authors are advised to carefully review and correct all occurrences throughout the manuscript.
2. The formatting of the email addresses in the affiliation section should follow the journal's guidelines.
3. In the introduction, it is stated on line 77 that reference 21 discusses a study on the antinociceptive activity of Ximenia americana extract. Additionally, on line 83, it is mentioned that reference 29 explores the antinociceptive effects of caffeic acid. Given this context, could the authors clarify the rationale for conducting this research again? Furthermore, could they elaborate on any new or distinctive findings that set this study apart from previous research?
4. In the results section, Figure 1 is missing a caption (1A and 1B).
5. The captions for Figures 3 and 4 appear to be the same. Kindly review and confirm their correctness.
6. In Figure 4, the abbreviation "EHXA" is used, but its full form has not been provided anywhere in the manuscript. The authors are requested to include the full form of this abbreviation.
7. In section 2.4, Analysis of signaling pathways underlying the analgesic effect of HEXA and CA, some experiments involve pretreatment with certain substances. For example, in section 2.4.3, Opioid pathway, pretreatment with naloxone is mentioned. The authors should provide additional details explaining the rationale for using these pretreatments.
8. There are two different formats used for writing "K+ ATP" in the manuscript. The authors are requested to ensure consistency throughout the manuscript and use the correct format in accordance with standard writing conventions.
9. On page 15, line 267, Figure 2 is mentioned again, even though it has already been discussed. The authors should review and verify the accuracy.
10. The Discussion section is engaging and effectively critiques the experimental results in comparison to previous studies. However, some paragraphs lack seamless transitions. For instance, paragraphs 4 and 5 do not flow cohesively. The authors are encouraged to improve the connections between these paragraphs to enhance the overall coherence of the section.
11. In the Materials and Methods section, it is mentioned that the plant extract used in this study, HEXA, was described in the authors’ previous work referenced as publication 20 from 2018. Could the authors clarify whether the extract used in this study is the same as the one from the earlier research? If it is the same extract, how can the authors ensure that the quality of the extract has not deteriorated over time? On the other hand, if a new extract was used, the authors should provide additional details about its source, such as the collection date, the location of the plant material used for extraction, and any differences in the handling or preparation processes.
12. In the Conclusion section, the abbreviation "SNC" is mentioned on line 524, but its full form has not been introduced earlier in the manuscript. The authors are advised to provide the full form of this abbreviation when it is first mentioned.
13. Some of the references cited in the manuscript are over 10 years old. If possible, the authors should consider replacing them with more recent references, preferably published within the last 10 years, to ensure the manuscript reflects the latest advancements in the field.
There are instances of typographical errors and deviations from proper English grammar throughout the manuscript. Therefore, the authors are strongly encouraged to thoroughly proofread and review the grammar and language of the manuscript before submitting the revised version.
Author Response
- The scientific names used in the manuscript should adhere to the proper conventions for writing scientific names as per international standards. For example, the genus and specific epithet should be italicized, the complete scientific name should include the author's name, and when mentioned subsequently, the genus name should be abbreviated to its initial, capitalized, and followed by a full stop. In this manuscript, some instances of scientific names do not comply with these conventions. Therefore, the authors are advised to carefully review and correct all occurrences throughout the manuscript.
Response: Thanks for the comment. All scientific names have been reviewed and corrected in accordance with international written standards
- The formatting of the email addresses in the affiliation section should follow the journal's guidelines.
Response: The email addresses have been corrected, I appreciate the comment.
- In the introduction, it is stated on line 77 that reference 21 discusses a study on the antinociceptive activity of Ximenia americana extract. Additionally, on line 83, it is mentioned that reference 29 explores the antinociceptive effects of caffeic acid. Given this context, could the authors clarify the rationale for conducting this research again? Furthermore, could they elaborate on any new or distinctive findings that set this study apart from previous research?
Response: Thank you for your comment. This study's main contribution was to elucidate the mechanism of action underlying the antinociceptive effect by investigating the associated signaling pathways for HEXA and caffeic acid. This work also adds new knowledge to the existing literature by providing further insights into the potential effects of caffeic acid.
- In the results section, Figure 1 is missing a caption (1A and 1B).
Response: Thank you for the comment. Figures 1A and 1B are described in the results section; however, this figure was moved to supplementary material at the suggestion of another reviewer.
- The captions for Figures 3 and 4 appear to be the same. Kindly review and confirm their correctness.
Response: Thank you for the comment. The captions of all figures were reviewed. The Figure 2 represented by the formalin test and Figure 3 by the participation of the nociceptive pathways.
- In Figure 4, the abbreviation "EHXA" is used, but its full form has not been provided anywhere in the manuscript. The authors are requested to include the full form of this abbreviation.
Response: Thank you for the comment. The description of the abbreviation HEXA has been added to the general objective, at the end of the introduction.
- In section 2.4, Analysis of signaling pathways underlying the analgesic effect of HEXA and CA, some experiments involve pretreatment with certain substances. For example, in section 2.4.3, Opioid pathway, pretreatment with naloxone is mentioned. The authors should provide additional details explaining the rationale for using these pretreatments.
Response: Thank you for the comment. The description of K+ATP has been standardized throughout the text.
- There are two different formats used for writing "K+ ATP" in the manuscript. The authors are requested to ensure consistency throughout the manuscript and use the correct format in accordance with standard writing conventions.
Response: Thank you for the comment. The repeated mention of Figure 1 has been replaced by Figure 4.
- On page 15, line 267, Figure 2 is mentioned again, even though it has already been discussed. The authors should review and verify the accuracy.
Response: Done. Thank you for the comment.
- The Discussion section is engaging and effectively critiques the experimental results in comparison to previous studies. However, some paragraphs lack seamless transitions. For instance, paragraphs 4 and 5 do not flow cohesively. The authors are encouraged to improve the connections between these paragraphs to enhance the overall coherence of the section.
Response: Thanks for the comment. In the discussion section between paragraphs 4 and 5, a sentence was added to improve coherence between the topics covered.
- In the Materials and Methods section, it is mentioned that the plant extract used in this study, HEXA, was described in the authors’ previous work referenced as publication 20 from 2018. Could the authors clarify whether the extract used in this study is the same as the one from the earlier research? If it is the same extract, how can the authors ensure that the quality of the extract has not deteriorated over time? On the other hand, if a new extract was used, the authors should provide additional details about its source, such as the collection date, the location of the plant material used for extraction, and any differences in the handling or preparation processes.
Response: Thanks for the comment. The extract from the present study belongs to the sample of previously published research (DA SILVA, Bruno Anderson Fernandes et al. HPLC profile and antiedematogenic activity of Ximenia americana L. (Olacaceae) in mice models of skin inflammation. Food and Chemical Toxicology, v. 119, p. 199-205, 2018). The same extract utilized was stored previously in lyophilized powder form under vacuum conditions and a new chemical analysis did not show significant alterations in the composition.
- In the Conclusion section, the abbreviation "SNC" is mentioned on line 524, but its full form has not been introduced earlier in the manuscript. The authors are advised to provide the full form of this abbreviation when it is first mentioned.
Response: Thank you for the comment. The description of the acronym referring to the central nerve system (CNS) has been added.
- Some of the references cited in the manuscript are over 10 years old. If possible, the authors should consider replacing them with more recent references, preferably published within the last 10 years, to ensure the manuscript reflects the latest advancements in the field.
Response: Thanks for the comment. The references were updated; however, some references that are over 10 years old have been kept because they describe classical methods for the field of inflammation and nociception.
- There are instances of typographical errors and deviations from proper English grammar throughout the manuscript. Therefore, the authors are strongly encouraged to thoroughly proofread and review the grammar and language of the manuscript before submitting the revised version.
Response: Thank you for the comment. The corrections were made. Spacing, punctuation marks, grammar, and spelling errors were reviewed by a native professional of English assistance
Round 2
Reviewer 4 Report
Comments and Suggestions for Authors
Based on the researcher's revisions, responses to questions, and well-executed engagement, I consider that this manuscript is acceptable for publication in the journal.